# Optimization of Conditions for Production of Soluble *E. coli* Poly(A)-Polymerase for Biotechnological Applications

**DOI:** 10.3390/biology14010048

**Published:** 2025-01-09

**Authors:** Igor P. Oscorbin, Maria S. Kunova, Maxim L. Filipenko

**Affiliations:** 1Institute of Chemical Biology and Fundamental Medicine, Siberian Branch of the Russian Academy of Sciences (ICBFM SB RAS), 8, Lavrentiev Avenue, Novosibirsk 630090, Russia; uwow@gmail.com (M.S.K.); mlfilipenko@gmail.com (M.L.F.); 2Novosibirsk State University, Novosibirsk 630090, Russia

**Keywords:** *pcnB*, poly(A) polymerase, PAP 1, polyadenylation, *E. coli*, recombinant protein, protein expression

## Abstract

Poly(A) polymerase (PAP 1) is the primary enzyme responsible for synthesizing poly(A) tails on RNA molecules, which signal RNA degradation in *Escherichia coli*. In vitro, PAP 1 is utilized for preparing RNA-seq libraries and producing mRNA vaccines. However, the toxicity of *E. coli* PAP 1 and its instability in low-salt buffers complicate its expression and purification. Here, we optimized the production of recombinant PAP 1 using seven *E. coli* strains with different genetic backgrounds. Cell density, overall protein yield, solubility, and enzymatic activity were used as criteria to evaluate the efficacy of PAP 1 production. BL21 (DE3) pLysS achieved the best balance of cell density, total yield, solubility, and specific activity of PAP 1. In Rosetta 2 (DE3) and Rosetta Blue (DE3) hosting the pRARE plasmid, PAP 1 was predominantly insoluble, likely due to excessive translation efficiency caused by the presence of rare tRNAs. Induction temperatures more than 18 °C also decreased PAP 1 solubility. Interestingly, PAP 1 accumulation positively correlated with the copy number of the PAP 1-encoding plasmid, suggesting its potential as a surrogate marker for PAP 1 activity. These findings provide valuable insights for optimizing the production of *E. coli* PAP 1 for biotechnological applications.

## 1. Introduction

Polyadenylation is a well-known nuclear process in eukaryotic cells, but polyadenylated RNA molecules have also been found in bacteria. While the relationship between polyadenylation and RNA degradation has been investigated, the exact details of this cellular process remain unclear [1]. At least three bacterial enzymes—polynucleotidylphosphorylase (PNPase) [2], RNase PH [3], and poly(A) polymerase (PAP)—are known to synthesize polyadenylic tails in a template-independent manner. Among these, poly(A) polymerase is primarily responsible for most polyadenylation under normal conditions, and its malfunction leads to a significant reduction in polyadenylated RNA molecules [4,5,6,7]. PAP belongs to the nucleotidyltransferase superfamily, along with tRNA nucleotidyltransferases, which add CCA tails to tRNA molecules during tRNA maturation [8].

Despite its discovery in the early 1960s, only a few bacterial PAPs have been purified and biochemically characterized. These include two enzymes from *E. coli* [9,10], one from *Geobacter sulfurreducens* [3], and another from *Pseudomonas putida* [11]. However, the *Pseudomonas putida* PAP has not been cloned and might represent a different enzyme capable of synthesizing poly(A) tails. *E. coli* PAP 1 is the most extensively studied, with determined optimal conditions in terms of the temperature, cofactors, salt concentration, and pH [12,13]. The *pcnB* gene encoding *E. coli* PAP 1 was identified as a determinant of the plasmid copy number in ColE1 origins [4,14]. Later, Sarkar et al. defined the *pcnB* product as poly(A) polymerase [5], which had already been purified and characterized at that time. However, the biochemical properties of other bacterial PAPs have not been as thoroughly evaluated, limiting understanding of their cellular roles and potential practical applications. Recently, *pcnB* genes were identified in multiple bacterial species, but their respective PAPs were not characterized, and their properties remain unknown. This lack of empirical data also hampers the in silico prediction of *pcnB* genes and their differentiation from similar *cca* genes [15].

PAPs have gained attention as tools for polyadenylating in vitro synthesized RNA molecules, particularly libraries for RNAseq and mRNA for vaccine use. During the SARS-CoV-2 pandemic, mRNA vaccines demonstrated their potential to control viral spread, proving the technology’s promise for other applications such as anticancer vaccines [16,17]. As the development of mRNA vaccines accelerates, the demand for simplified and cost-effective reagents for their production increases. However, *E. coli* PAP 1 is known for its instability, including a tendency to lose activity due to slow aggregation in low-salt buffers [5,12]. Additionally, it contains eight cysteine residues, making it prone to remaining in insoluble fractions after cell lysis. Furthermore, *E. coli* tightly regulates PAP 1 expression because its overproduction is toxic to host cells, significantly reducing cell viability post-induction [6]. Together, these issues pose challenges when developing protocols to obtain soluble *E. coli* PAP 1 in high quantities.

One common approach to optimize recombinant protein production is selecting the most suitable host strain based on the protein’s solubility and yield [18,19]. Various *E. coli* strains have been engineered to improve recombinant protein production. Rosetta strains carry the plasmid pRARE, which encodes tRNAs that are rare in *E. coli*, enhancing the expression of heterologous proteins [20,21,22]. BL21 Gen-X is designed to increase expression yields, while SoluBL21 enhances the solubility of mammalian proteins [23,24,25]. Lemo21 (DE3) includes a plasmid encoding the T7 RNA polymerase inhibitor LysY, enabling expression tuning via the L-rhamnose concentration [26,27]. Shuffle T7 cells express DsbC disulfide bond isomerase to correct non-native disulfide bonds, along with an additional LacI protein to prevent promoter leakage [28,29].

In this study, we aimed to identify the most effective *E. coli* strain for producing recombinant *E. coli* PAP 1. Seven *E. coli* strains were tested to evaluate the enzyme’s yield and solubility. Additionally, we measured the ratio of plasmid carrying the *pcnB* gene to the host cell’s genomic DNA and compared these ratios with the yield, solubility, and activity of the recombinant enzyme. In the future, the production and solubility of PAP 1 can be further improved using several methods: co-expression with molecular chaperones; supplementation of the culture medium with additives such as sorbitol, ethanol, betaine, and L-arginine; a short temperature shock before the addition of IPTG to induce the synthesis of host heat shock proteins; and optimization of the lysis conditions, such as the ionic strength, detergents, and chaotropic agents.

## 2. Materials and Methods

### 2.1. Cloning of E. coli Poly(A) Polymerase

The coding sequence of *E. coli* PAP 1 (GenBank: M20574.1) was amplified using PcnB-Eco-F1/PcnB-Eco-R1 primers (Table 1) with NheI and XhoI restriction sites, allowing in-frame ligation into the pET36b vector (Novagen, Madison, WI, USA). PCR was carried out using genomic DNA of the *E. coli* strain XL1-Blue as a template. The resultant 1.6 kbp DNA fragment and pET36b vector were digested with NdeI and XhoI (SibEnzyme, Novosibirsk, Russia), ligated, and transformed into *E. coli* XL1-Blue cells according to the standard protocols [30]. The fidelity of the resulting recombinant plasmid, named pET-PAP-Eco, was confirmed by sequence analysis using the primers pET-F and pET-R (Table 1) using the Big Dye Terminator kit 3.1 (Applied Biosystems, Waltham, MA, USA) and the ABI 3730 genetic analyzer (Applied Biosystems, Waltham, MA, USA) in the laboratory of antimicrobial drugs at ICBFM SB RAS according to the manufacturer’s protocol.

### 2.2. Expression of E. coli Poly(A) Polymerase

A starter culture of *E. coli* strains (Table 2) harboring the plasmid pET-PAP-Eco was grown to OD_600_ = 0.6 in LB medium with 25 μg/mL kanamycin at 37 °C. For each strain, two cultures in 5 mL of medium each were prepared. After reaching the designated optical density, samples for PAGE were taken as controls before induction, IPTG was added to a final concentration 1 mM to induce the expression of PAP 1, and the cultures were divided into 1.5 mL subcultures that were subjected to ON incubation at 18, 25, or 37 °C. After induction, the OD_600_ was measured, the control samples after induction were sampled and cells from 1 mL were harvested by centrifugation at 4000× *g* and stored at −70 °C for a possible solubility test.

### 2.3. Solubility Test

To evaluate the solubility of PAP 1 *E. coli* in different *E. coli* strains, cell pellets from 1 mL night cultures were lysed in 200 μL of a lysis buffer containing 50 mM Tris-HCl, pH 8.0, 200 mM KCl, 1 mM EDTA, 5 mM beta-mercaptoethanol, 5% glycerol, 1 mM PMSF, 0.5% 3-(N,N-Dimethylmyristyl-ammonio)propanesulfonate, and 0.05% C7BzO. The cell pellets were resuspended in the lysis buffer following addition of lysozyme to 0.5 mg/mL and incubation for 1 h at 37 °C. All the incubated lysates were sonicated using a Cell Disruptor 200 (Branson Ultrasonics Corp., Brookfield, CT, USA) under the following conditions: 70 W for 1 min in pulsed mode with a 30% duty cycle, repeated twice. Samples from a single strain were lysed and sonicated within the same experiment to minimize variations in the lysis time between samples. The sonicated lysates were centrifuged at 20,000× *g* for 15 min. Soluble fractions were transferred into new tubes, and insoluble pellets were resuspended in 200 μL of the lysis buffer. Both soluble and insoluble fractions were analyzed using SDS-PAGE in 12.5% acrylamide gel, and the resulting electropherograms were quantified using the ImageLab 6.1 software (Bio-Rad, Hercules, CA, USA).

### 2.4. Polyadenylation Assay

The specific activity of poly(A) polymerase was analyzed using elongation of the fluorescently labeled oligo(r)A_20_ oligonucleotide. The reaction mixes contained a volume of 10 μL 1× reaction buffer for PAP 1 (50 mM Tris-HCl, 250 mM NaCl, 10 mM MgCl_2_, pH 8.0), 1 mM ATP, 10 pmol of (r)A20-oligonucleotide, and a specified below amount of bacterial lysates after induction. The reactions were started by the addition of the enzyme and immediately transferred to a preheated thermocycler, followed by an incubation for 30 min at 37 °C. After incubation, the reactions were quenched by the addition of 10 μL of formamide and denatured by heating for 5 min at 95 °C. The reaction products were analyzed using denaturing PAGE in a 18% acrylamide gel with 7 M urea and quantified using the ImageLab software (Bio-Rad, Hercules, CA, USA).

### 2.5. Quantitative PCR

Quantitative PCR reactions were performed in a 20 µL volume containing 65 mM Tris-HCl, pH 8.9, 24 mM (NH_4_)_2_SO_4_, 0.05% Tween-20, 3 mM MgSO_4_, 0.2 mM dNTPs, 600 nM primers Eco20-F, Eco20-R, CANP-F, CANP-R, 100 nM TaqMan probes Eco20-P, CANP-P (Table 1) and 1 U of Taq-polymerase (Biosan, Novosibirsk, Russia). Amplification was carried out in the CFX96 Real-Time PCR Detection System (Bio-Rad, Hercules, CA, USA) according to the following program: 95 °C for 3 min, followed by 45 cycles of 95 °C for 10 s, and 60 °C for 40 s, with a collection of fluorescent signals in the FAM and ROX channels. The acquired data were analyzed with CFX Manager 3.1 software (Bio-Rad, Hercules, CA, USA).

### 2.6. Data Analysis

The relative amount of the polyadenylated substrate was calculated as follows—Ratio = I(elongated products)/I(substrate) + I(elongated products))—and used to estimate by a linear regression model a lysate amount necessary to elongate 50% of the substrate initially added to the reaction. These computed lysate volumes served to assess the PAP 1 activity. To account for a final cell density, the computed lysate volumes were divided based on the respective OD_600_ values. 

The correlations between variables (induction temperature, OD_600_, PAP 1 level, etc.) were computed by Spearman’s rank correlation test with a two-tailed *p*-value and 95% confidence intervals and visualized in https://www.chiplot.online (accessed on 29 December 2024). All the calculations were performed in the GraphPad Prism 8.0.1 software (Insight Venture Management, New York, NY, USA).

## 3. Results

### 3.1. Expression of E. coli PAP 1

To evaluate the production of recombinant PAP 1 in *E. coli* cells, we cloned the PAP 1 coding sequence into the expression vector pET36b, placing it under the control of the lac promoter, thereby generating the plasmid pET36b-PAP1-Eco. The pET36b vector enables tight regulation of recombinant protein expression via the recombinant LacI protein encoded by the plasmid. Additionally, the vector’s antibiotic resistance gene and PAP 1 coding sequence are located in separate transcriptional units, preventing unintended PAP 1 expression due to the synthesis of a polycistronic RNA from the antibiotic resistance gene promoter.

Seven *E. coli* strains were tested for PAP 1 expression, including the commonly used BL21 (DE3) pLysS strain, its derivatives optimized for recombinant protein production, and two K-12 strain derivatives, Rosetta Blue (DE3) and Shuffle T7 (Table 2). This selection allowed us to consider the impact of different genetic backgrounds on PAP 1 expression. After transformation with pET36b-PAP1-Eco, *E. coli* cells were grown to an OD_600_ of 0.6, and PAP 1 synthesis was induced by the addition of IPTG to a final concentration of 1 mM. The induced cultures were divided into three equal portions and incubated overnight at 18 °C, 25 °C, and 37 °C, respectively. Following induction, the optical density of the overnight cultures was measured, and the relative amount of recombinant PAP 1 was evaluated using SDS-PAGE (Figure 1).

After PAP 1 induction, the highest optical density (OD_600_) was observed in the Rosetta Blue (DE3) strain, exceeding three. In contrast, the BL21 Gen-X strain demonstrated the lowest OD_600_, which did not exceed one across all the induction temperatures. Similarly, SoluBL21 showed low growth, except for the cultures incubated at 37 °C, which achieved an OD_600_ of 2.6. These findings align with the manufacturer’s notes, which indicate a reduced growth rate for these strains. For the remaining *E. coli* strains, the cell density was moderate, with the OD_600_ values ranging from 1.5 to 2.3. Notably, only three strains—BL21 (DE3) pLysS, SoluBL21, and Shuffle T7—exhibited higher cell densities at 37 °C compared to the other temperatures. For the other strains, the OD_600_ values showed no consistent correlation with the induction temperature (Figure 2).

The highest PAP 1 levels were observed in the Shuffle T7 and BL21 (DE3) pLysS strains, reaching 1–1.5% of the total protein, while the lowest levels were found in BL21 Gen-X and SoluBL21, at 0.18–0.24%. Interestingly, for strains showing increased OD_600_ at 37 °C, the PAP 1 levels at this temperature were either lower than or equal to those at 18 °C and 25 °C. Thus, whether analyzed individually or collectively, the PAP 1 levels did not show a positive correlation with the cell optical density after induction.

### 3.2. Solubility of E. coli PAP 1

After measuring the bulk PAP 1 level after induction, we assessed the solubility of the recombinant protein. For that, the cell pellets were lysed in a buffer containing 200 mM KCl, two detergents (3-(N,N-Dimethylmyristyl-ammonio)propanesulfonate, 0.05% C7BzO) reported to be efficient reagents for fast cell disruption and lysozyme. The insoluble and soluble fractions were separated by centrifugation and the protein compositions were analyzed using SDS-PAGE (Figure 3).

Unlike the overall PAP 1 levels, the solubility of PAP 1 showed a moderate negative correlation with the induction temperature. For all the tested *E. coli* strains, higher induction temperatures resulted in decreased solubility. In most strains, PAP 1 was nearly 100% soluble when expressed at 18 °C, whereas only 42–65% of the recombinant enzyme remained soluble at 37 °C. An exception to this trend was Lemo21 (DE3), where no soluble PAP 1 was detected at 37 °C. A possible reason why PAP 1 was insoluble in Lemo21 (DE3) could be the rapid production of the recombinant protein, leading to its accumulation in inclusion bodies. This strain is designed for the tunable expression of recombinant proteins regulated by the L-rhamnose level. In this study, we did not adjust the expression of PAP 1 by adding L-rhamnose, which might have resulted in the rapid overproduction of PAP 1. Additionally, in both Rosetta strains (Rosetta 2 (DE3) and Rosetta Blue (DE3)), the PAP 1 solubility was notably lower, with a maximum solubility of approximately 20% at 18 °C. These strains originate from different lineages but share the pRARE plasmid, which encodes tRNAs that are rare in *E. coli*. 

### 3.3. Specific Activity of E. coli PAP 1

Since protein bands on SDS-PAGE do not always accurately reflect the quantity of active enzymes, we further tested the poly(A) polymerase activity in the samples after PAP 1 expression. *E. coli* cells possess at least three enzymes capable of polyadenylating RNA: PAP 1, PAP 2, and PNPase. The activity of these enzymes in crude lysates after recombinant PAP 1 expression could distort the results of polyadenylation assays. To account for the host polyadenylation activity, we used control cultures of all the strains that did not contain the pET-PAP-Eco plasmid, which could produce recombinant PAP 1 due to promoter leakage. All the control cultures were prepared identically to those expressing recombinant PAP 1. No elongation of the (r)A20-oligonucleotide was observed with any control sample, indicating a low basal level of polyadenylation enzymes (an example is provided in Appendix A). Therefore, the host enzymes did not distort the results of the polyadenylation activity in the crude lysates. For the PAP 1 activity assay, a fluorescently labeled 20-mer A oligonucleotide was incubated with varying volumes of cell lysates in a reaction buffer optimized for *E. coli* PAP 1. To minimize the potential inhibition by cellular components, the lysate volumes were titrated in the range of 0.015–1 μL per reaction.

The elongated reaction products were quantified, and the lysate volumes required to polyadenylate 50% of the substrate were calculated. These computed lysate volumes served as a measure of the PAP 1 activity in each sample. To account for differences in the cell density across the strains, the calculated lysate volumes were normalized to the respective OD_600_ values. Higher computed lysate volumes indicated lower PAP 1 activity in the corresponding samples. The results of the activity analysis are presented in Figure 4.

A weak positive correlation was observed between the PAP 1 activity and the PAP 1 solubility: the more PAP 1 that remained in the soluble fraction, the more active it was. However, no correlation was found between the PAP 1 activity and the expression temperature, indicating that PAP 1 remained active even in the insoluble state when the induction temperature was increased. A moderate positive association was noted between the PAP 1 activity and the amount of DNA in the cells, whether genomic or plasmid. Note that the activity in Figure 2 is given in computed lysate volumes; thus, the positive correlation between the Cq values and the activity on the heatmap should be understood as lower lysate volumes (meaning higher activity) correlating with lower Cq values. The increased PAP 1 activity positively correlated with the plasmid copy number, suggesting that relaxation of plasmid replication occurred due to the accelerated degradation of antisense RNA-1, which normally inhibits priming at ColE1 origins.

### 3.4. Relation Between Plasmid Copy Number and Level of Recombinant PAP 1

The PAP 1 gene, *pcnB*, was identified as a locus involved in the regulation of the plasmid numbers in *E. coli* cells. Specifically, PAP 1 polyadenylates a small RNA1 transcript, which negatively regulates the replication of plasmids with a ColE1-based origin. Therefore, we hypothesized that there could be a link between the PAP 1 levels and the copy number of the plasmid encoding PAP 1, i.e., high PAP 1 expression might lead to an increase in plasmid numbers in host cells. To test this hypothesis, we measured the amounts of *E. coli* genomic DNA and the pET-PAP-Eco plasmid using qPCR. The results are presented as Cq values in Figure 5a. Notably, the Cq values are inversely correlated with the amount of the respective DNA template. Since Cq values are logarithmic, they serve as relative indicators of the DNA template quantity in the analyzed sample.

Unlike the optical density, no correlation was found between the expression temperature and the Cq values for genomic DNA. Thus, qPCR did not detect any temperature-dependent changes in the accumulation of cellular DNA. However, the situation was different for the pET-PAP-Eco plasmid, where the plasmid numbers increased at 37 °C compared to 25 °C in the Lemo21 (DE3), SoluBL21, Rosetta Blue (DE3), and Shuffle T7 strains. In contrast, no significant trend was observed for BL21 Gen-X.

To further investigate potential changes in the DNA amounts, we calculated the difference between the Cq values for the genomic DNA and the pET-PAP-Eco plasmid. This difference served as a rough measure of the ratio between the amounts of genomic DNA and the plasmid, assuming similar qPCR efficiency for both targets. The calculated delta Cq values are presented in Figure 5b, showing a moderate positive correlation between the expression temperature and the pET-PAP-Eco plasmid copy numbers.

The Cq and ΔCq values for both the genomic DNA and the pET-PAP-Eco plasmid moderately correlated with the PAP 1 levels. Surprisingly, a strong negative correlation was observed between the ΔCq values and the PAP 1 solubility, while the solubility was not associated with the plasmid’s Cq values and was positively correlated with the amount of genomic DNA. This suggests that the higher insolubility of PAP 1 was associated with higher plasmid copy numbers. However, the PAP 1 activity positively correlated with the amounts of genomic DNA, plasmid, and plasmid copy number.

## 4. Discussion

The rapid progress in the development and commercialization of mRNA vaccines has highlighted the importance of enzymes required for mRNA processing, particularly when the enzymatic method is chosen. *E. coli* PAP 1 is one such enzyme, as polyadenylation of mRNA molecules is essential for their stability in eukaryotic cells. Therefore, the efficient production of *E. coli* PAP 1, along with other recombinant poly(A) polymerases, is a crucial step in mRNA production. However, PAP 1 is toxic to host cells, prone to aggregation, and loses activity when stored in low-salt conditions. In *E. coli*, PAP 1 production is tightly regulated to prevent negative impacts on the cell viability and growth rate, making the expression of recombinant PAP 1 a challenging task. Notably, *E. coli* PAP 1 is a difficult protein for recombinant production due to its high pI (9.67), its high charge at pH 7.0 (16.51), and the presence of four cysteine residues.

In this study, we examined the PAP 1 expression in seven *E. coli* strains known for producing “difficult” recombinant proteins, including toxic, membrane-bound, insoluble proteins, and proteins with a high number of disulfide bonds. BL21 (DE3) pLysS was chosen as a reference strain because it is commonly used for recombinant protein expression. BL21 Gen-X and SoluBL21 (DE3) are optimized for producing mammalian and soluble proteins, respectively. LemoBL21 (DE3) offers tunable expression of recombinant proteins regulated by L-rhamnose concentration. Shuffle T7 is a K-12-derived strain with a DsbC chaperone to correct non-native disulfide bonds and additional LacI to prevent lac promoter leakage. Rosetta 2 (DE3) is a BL21 (DE3) derivative with a pRARE2 plasmid, providing tRNAs to suppress the codon bias for eukaryotic proteins. Rosetta Blue (DE3) is a K-12-based strain with the same pRARE plasmid and a *lacI^q^* mutation, similar to Shuffle T7. Thus, five strains were from the B lineage of *E. coli*, while two originated from the K-12 lineage. The B lineage strains lack the cytoplasmic Lon and membrane OmpT proteases, but it was unknown whether these enzymatic activities affect PAP 1 production in *E. coli*. However, no differences in the optical density, relative amount, or solubility were observed between the B and K-12 strains.

Previous studies have shown that *E. coli* PAP 1 is highly toxic, causing cell death within 30 min of IPTG induction [6]. However, in our experiment, all the strains except for the slow-growing BL21 Gen-X (DE3) and SoluBL21 (DE3) reached an OD_600_ of 1.5–3.5, and PAP 1 production was directly associated with the final optical density. While we did not assess the cell viability post-induction, these results seem to contradict the reported toxicity of PAP 1. This discrepancy may be explained by the delayed toxic effects of PAP 1, where high enzyme production does not cause immediate cell death.

Both Rosetta strains, derived from different lineages but sharing the pRARE plasmid, exhibited the lowest solubility of PAP 1. It remains unclear whether the pRARE plasmid or other unknown factors contributed to PAP 1 aggregation into inclusion bodies. One possible explanation is that the higher translation rate of PAP 1, facilitated by the rare tRNAs encoded by the pRARE plasmid, may lead to aggregation. This hypothesis is supported by several observations: first, rare codons generally have a minimal impact on protein expression in *E. coli* [31]; second, an increased translation rate of heterologous proteins in *E. coli* strains with the pRIL plasmid encoding rare tRNAs has been associated with reduced solubility [32]; and third, a high percentage of arginine correlates with lower expression and solubility [33]. The nucleotide sequence of PAP 1 contains 18 rare codons (mainly encoding threonine) per 466 codons, as well as an unusually high proportion of arginine residues (11.4%, or 53 residues). However, we did not directly test this hypothesis by comparing the PAP 1 expression with and without the pRARE plasmid. In other strains, the solubility depended on the expression temperature without any notable anomalies.

The activity and level of PAP 1 were lowest in BL21 Gen-X (DE3) and SoluBL21 (DE3), the slowest-growing strains. We speculated that the low PAP 1 production in these strains was due to its toxicity. Other strains, which grew faster, produced more cells expressing recombinant PAP 1. Therefore, if these *E. coli* cells lost viability due to PAP 1 accumulation, many of the dead cells would still contain the active enzyme. Conversely, the slower-growing strains did not produce a substantial quantity of PAP 1 due to the limited time for expression, resulting in a less productive culture.

The lack of association between the solubility and the specific activity of PAP 1 can be attributed to several potential explanations. First, the enzyme may retain its activity even in an insoluble state. Second, PAP 1 might gradually dissolve from inclusion bodies when exposed to the high ionic strength of the PAP 1 reaction buffer. This second hypothesis is supported by PAP 1’s known tendency to aggregate under low-salt conditions. However, whether these assumptions are valid or if other factors influence PAP 1 activity remains unclear.

A fruitful strategy to increase the yield, solubility, and activity of recombinant proteins in *E. coli* cells is supplementing the culture medium with various chemicals, such as ethanol [34], sorbitol, betaine [35], arginine [36], and glycerol [37]. These substances alter the cellular metabolism and often increase the expression of chaperones, thus facilitating the synthesis of heterologous proteins. The main goal of this study was to compare the ability of different *E. coli* strains to produce soluble and active *E. coli* PAP 1, which did not assume the use of additional chemical substances in the culture medium. Our results demonstrated that the host strain has a significant impact on the efficacy of PAP 1 production, and proprietary *E. coli* strains are not always the best choice for this task. While seemingly time-consuming and cumbersome, systematic preliminary experiments can save both funds and time if they demonstrate that freely available *E. coli* strains express the same amount of recombinant protein as commercial strains. Further optimization, specifically co-expression with heat shock proteins and supplementation of the culture medium with various chemicals, could additionally increase PAP 1 production, as shown for other proteins. However, in large-scale production, additional chemicals can unexpectedly increase the cost of fermentation, not only because of the cost of the substances themselves but also due to the need for proper waste disposal. The use of heat shock proteins may also be limited by their negative impact on *E. coli* propagation as an additional physiological burden or by their cost if commercially available proteins are used. Thus, the resulting production cost needs to be carefully considered before implementing any additives into the manufacturing process. In light of this, strain optimization appears to be one of the most economical approaches, as it only needs to be performed once and does not require additional expenditures if a freely available strain demonstrates an acceptable production rate.

Intriguing results emerged when we explored the potential correlations between the DNA amounts, plasmid copy numbers, and PAP 1 properties. Higher PAP 1 activity corresponded with an increased plasmid DNA quantity and a higher plasmid copy number. This observation aligns with PAP 1’s role in degrading RNA-1, which negatively regulates replication of ColE1-based plasmids [4]. However, the strong negative association between the solubility and the plasmid copy number was unexpected. We speculate that the rapid accumulation of PAP 1 facilitated efficient plasmid propagation, leading to its aggregation and incorporation into inclusion bodies. Thus, despite the relatively low solubility of PAP 1 in the Rosetta strains, its activity was still evident through changes in the plasmid copy numbers. The increased genomic DNA content observed presented another challenge, as no direct link between PAP 1, polyadenylation, and *E. coli* genome replication has been identified. This elevated DNA content may result from other cellular processes not accounted for in this study.

To summarize, we evaluated the ability of several *E. coli* strains to produce recombinant *E. coli* PAP 1 at different induction temperatures by analyzing the enzyme’s overall level, solubility, and specific activity. Here, we focused solely on comparing different *E. coli* strains for the production of recombinant *E. coli* PAP 1. Other methods commonly used to increase the yield and solubility of recombinant proteins, such as co-expression with molecular chaperones, supplementation of the culture medium with additives, induction of host heat shock proteins, and optimization of lysis conditions (e.g., ionic strength, detergents, and chaotropic agents), could also be beneficial but remain subjects for possible future studies.

## 5. Conclusions

In this study, we evaluated the production of recombinant PAP 1 across seven different *E. coli* strains. The BL21 (DE3) pLysS strain demonstrated the optimal balance of cell density, overall PAP 1 yield, solubility, and specific activity. In contrast, the Rosetta strains displayed the lowest solubility of the recombinant enzyme, potentially due to excessively high translation efficiency. Induction temperatures above 18 °C led to the increased insolubility of PAP 1. Notably, *E. coli* PAP 1 accumulation was associated with an increase in the plasmid copy number encoding the enzyme, suggesting its potential use as a surrogate marker of PAP 1 activity.

## Figures and Tables

**Figure 1 biology-14-00048-f001:**
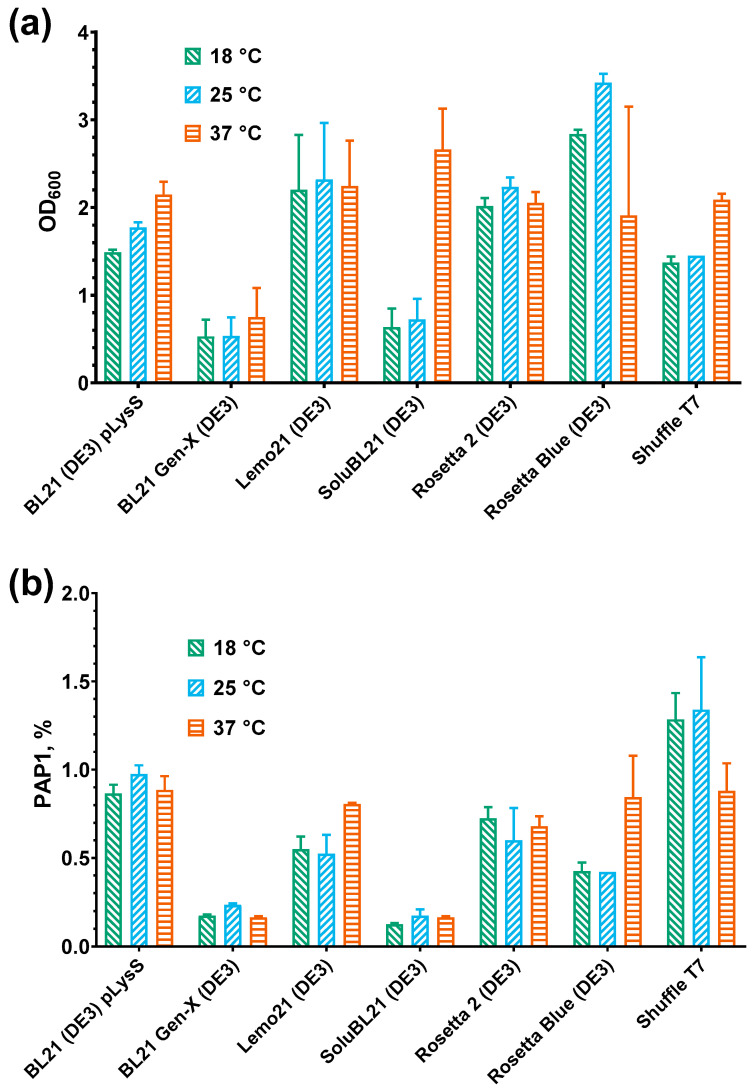
Optical density and level of PAP 1 after PAP 1 expression. (**a**) Optical density of *E. coli* cultures after overnight expression of PAP 1 at different temperatures (18, 25, 37°). Y-axis marks the OD_600_ values; X-axis represents the strains. (**b**) Relative quantity of PAP 1. The PAP 1 percent was calculated after densitometry as a relation of a PAP 1 band to all the protein bands in the respective lane. Y-axis designates the recombinant PAP 1 percent; X-axis designates the strains. The expression temperature is designated by the color, as specified in the legend. All the experiments were triplicated; the error bars demonstrate the SD.

**Figure 2 biology-14-00048-f002:**
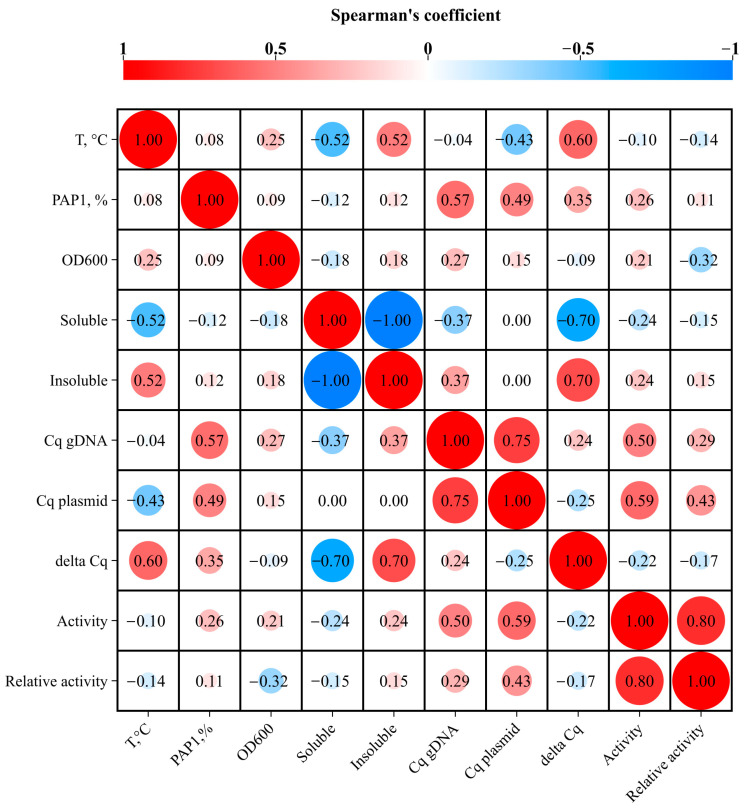
A heatmap of the correlation between variables involved in PAP 1 induction. The heatmap made in ChiPlot (https://www.chiplot.online (accessed on 29 December 2024)) represents the Spearman’s rank coefficients for each pair of variables. The coefficients are indicated in each cell. Red color means positive correlation; blue color designates negative correlation. The size of the circles inside the cells represents the size of the respective correlation.

**Figure 3 biology-14-00048-f003:**
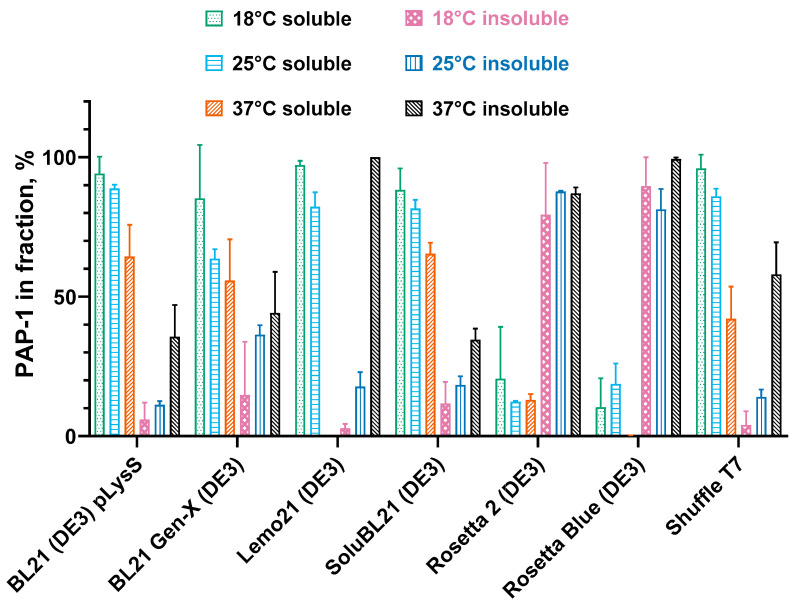
Solubility of recombinant PAP 1 after expression in different strains. The cell pellets after PAP 1 expression were lysed, followed by centrifugation to divide the soluble and insoluble fractions, which were analyzed by SDS-PAGE. The PAP 1 percent in each fraction was calculated after densitometry as a relation of a PAP 1 band to all the protein bands in the respective lane. Y-axis marks the PAP 1 percent in the soluble or insoluble fractions; X-axis represents the strains. The expression temperature and protein fraction are designated by the color, as specified in the legend. All the experiments were triplicated; the error bars demonstrate the SD.

**Figure 4 biology-14-00048-f004:**
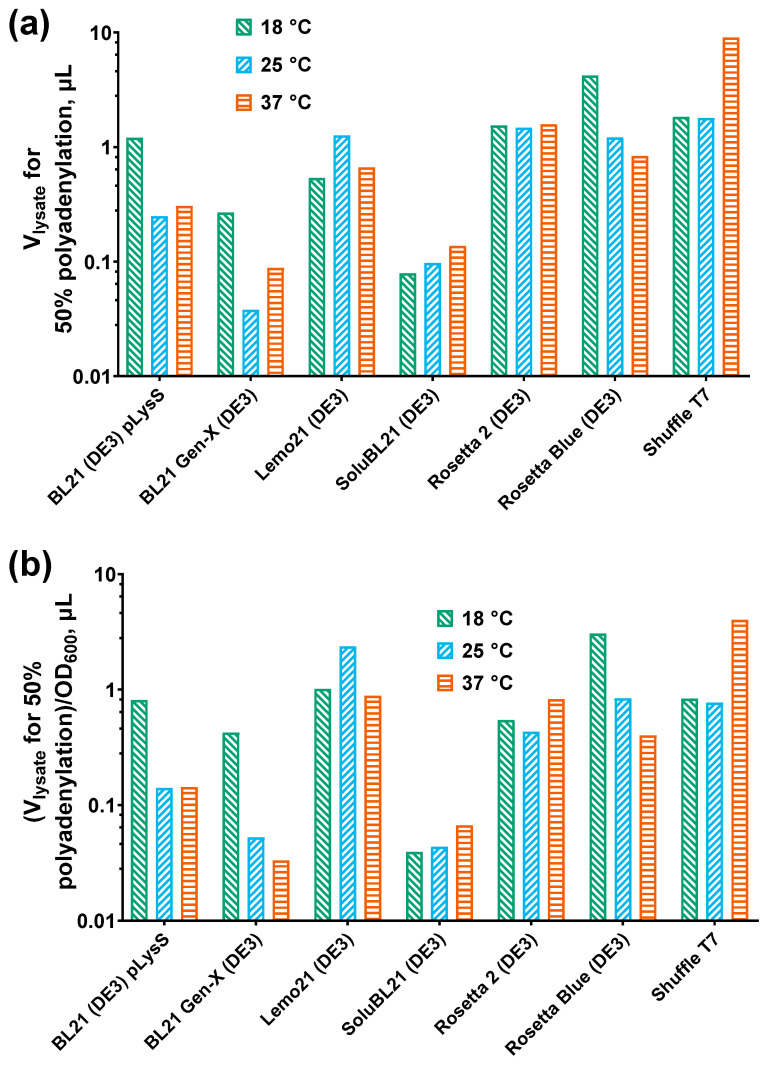
Specific activity of PAP 1. The specific activity of PAP 1 was measured using a fluorescently labeled (r)A_20_ substrate that was incubated with lysates after expression of recombinant *E. coli* PAP 1. The reaction products were analyzed in denaturing PAGE and quantified using ImageJ. (**a**) Computed lysate volumes for polyadenylation of 50% of the substrate. Y-axis marks the computed lysate volumes necessary to polyadenylate 50% of the substrate. X-axis represents the strains. (**b**) Computed lysate volumes adjusted to the respective OD_600_. Y-axis marks the computed lysate volumes necessary to polyadenylate 50% of the substrate adjusted to the respective OD_600_ values. X-axis represents the strains. The expression temperature is designated by the color, as specified in the legend. The PAP 1 activity was measured using the elongation of the fluorescently labeled oligo(r)A_20_ substrate.

**Figure 5 biology-14-00048-f005:**
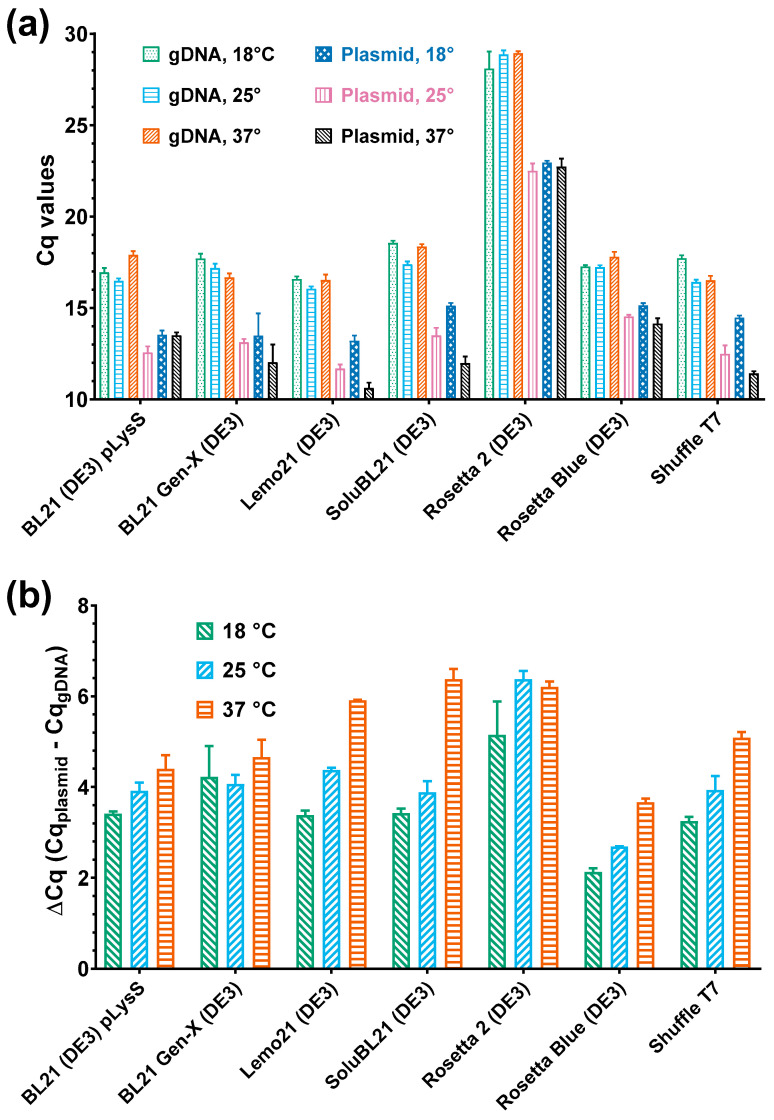
Amounts of *E. coli* genomic DNA and PAP 1-encoding plasmid after PAP 1 expression. The DNA amounts were evaluated using quantitative PCR and the cultures after expression as templates; Cq values reflected the concentrations of gDNA and plasmid. (**a**) Cq values of the qPCR for the quantification of the *E. coli* genomic DNA and PAP 1-encoding plasmid pET-PAP-Eco. Y-axis marks the Cq values; X-axis represents the strains. (**b**) Relative quantity of pET-PAP-Eco. Y-axis designates the difference between the Cq for the plasmid and the gDNA; X-axis designates the strains. All the experiments were triplicated; the error bars demonstrate the SD.

**Table 1 biology-14-00048-t001:** Oligonucleotide primers and probes.

Primer	5′-Sequence-3′	Restriction Endonuclease
PcnB-Eco-F1	CACACCATATGTACAGGCTCAATAAAGCGGG	NdeI
PcnB-Eco-R1	CACACCTCGAGCGACGTGGTGCGCG	XhoI
pET-F	CCTATAGTGAGTCGTATTAATTTC	
pET-R	CAACTCAGCTTCCTTTCGG	
Eco20-F	GTTCGAGAAGAAACTCGAAGC	
Eco20-R	CCAGATATACCTTAACAATCTTCAG	
Eco20-P	FAM-CGGTGCCAGATCGTCGCCC-BHQ1	
CANP-F	ATTCTTCTAATACCTGGAATGCTGT	
CANP-R	TTTATCCGTACTCCTGATGATGC	
CANP-P	ROX-CGGGGATCGCAGTGGTGAGT-BHQ2	

**Table 2 biology-14-00048-t002:** *E. coli* strains used for expression of PAP 1 *E. coli*.

Strain	Lineage	Genotype
BL21 (DE3) pLysS	BL21	F^−^ ompT gal dcm lon hsdSB(rB^−^ mB^−^) λ(DE3 [lacI lacUV5-T7p07 ind1 sam7 nin5]) [malB+]K-12(λS) pLysS[T7p20 orip15A](CmR)
BL21 Gen-X (DE3)	BL21	F^−^ ompT hsdSB (rB^−^ mB^−^) gal dcm (DE3)
Lemo21 (DE3)	BL21	fhuA2 [lon] ompT gal (λ DE3) [dcm] ∆hsdS/pLemo(CamR) λ DE3 = λ sBamHIo ∆EcoRI-B int::(lacI::PlacUV5::T7 gene1) i21 ∆nin5 pLemo = pACYC184-PrhaBAD-lysY
SoluBL21 (DE3)	BL21	F^−^ ompT hsdSB (rB^−^ mB^−^) gal dcm (DE3)
Shuffle T7	K-12	F′ lac, pro, lacIq/Δ(ara-leu)7697 araD139 fhuA2 lacZ::T7 gene1 Δ(phoA)PvuII phoR ahpC galE (or U) galK λatt::pNEB3-r1-cDsbC (SpecR, lacIq) ΔtrxB rpsL150(StrR) Δgor Δ(malF)3
Rosetta 2 (DE3)	BL21	F^−^ ompT hsdSB(rB^−^ mB^−^) gal dcm (DE3) pRARE2 (CamR)
Rosetta Blue (DE3)	K-12	endA1 hsdR17 (rK12^−^ mK12+) supE44 thi-1 recA1 gyrA96 relA1 lac (DE3) F′[proA+B+ lacIqZΔM15::Tn10] pRARE (CamR, TetR)

## Data Availability

Dataset available on request from the authors.

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
