# Peer review of "Optimization of Conditions for Production of Soluble E. coli Poly(A)-Polymerase for Biotechnological Applications"

_biology, 2025, doi:10.3390/biology14010048_

Round 1
Reviewer 1 Report
Comments and Suggestions for Authors
The manuscript titled "Optimization of Conditions for Production of Soluble E. coli PolyA-Polymerase" by Oscorbin et al. looks at how to improve the production of poly(A)-polymerase (PAP-1) in E. coli. This enzyme is essential for RNA polyadenylation and is used in RNA sequencing and mRNA vaccine development. The study compares the expression of PAP-1 in seven different E. coli strains to find the best conditions for producing the protein, making it soluble, and ensuring it works properly. The results emphasize how factors like the choice of E. coli strain, temperature during induction, and plasmid copy number are crucial for better protein production. These findings offer useful insights for improving PAP-1 production for use in biotechnology. While the experiment and results are solid, the manuscript could benefit from clearer statistical analysis and more discussion about how these results apply in real-world settings. I have a few comments and suggestions for improvement.
1. The manuscript contains minor grammatical issues that require attention. For instance, in line 11, the phrase 'PAP-1's toxicity and instability complicate its expression and purification' should be corrected to 'complicate.' Additionally, there is a typographical error in line 131 ('poply(A)') and line 122 (1 mM ЭДТА). Subject-verb agreement errors are observed throughout the manuscript. It is recommended that the authors review the text carefully to ensure grammatical accuracy and clarity.
2. The author thoroughly explained the method but did not include details regarding the sonication parameters used for cell lysis, such as duration, power settings, and on/off cycles. Were the same parameters applied to all samples? Was the lysis performed on individual samples or simultaneously? Given the significant impact of sonication parameters on cell lysis, protein folding, solubility, and activity, it would be helpful to clarify these aspects for a more comprehensive understanding of the experimental conditions.
3. It is well-established, and the author has also mentioned in two instances in the manuscript, that Lemo21(DE3) is designed to modulate the expression levels of potentially toxic or difficult-to-express proteins using a tunable T7 RNA polymerase system controlled by the rhaBAD promoter, which is regulated by L-rhamnose. Therefore, did the authors add L-rhamnose during protein expression induction? If so, this should be specified in the methods section. If not, the absence of L-rhamnose could be a contributing factor to the insolubility of the expressed protein.
4. The experimental methods are detailed but could be improved for clarity and reproducibility. Specify whether conditions such as MgCl₂ or other cofactors (ATP etc.) were tested during expression, given the enzyme's reliance on divalent metals for activity.
5. Some figures, such as the heatmap (Figure 2) and activity comparison (Figure 4), lack clear annotations and labels. For example, the color scale in the heatmap should be described in the legend, and key correlations should be highlighted. Additionally, Figure 4’s axes and legends could be better labeled to indicate the significance of the lysate volume adjustments for activity measurements.
6. In the experiment “Specific activity of E. coli PAP-1 (Figure 4),” calculating recombinant PAP-1 activity based on lysate volume, did the authors quantify the concentration of PAP-1 in each sample using methods such as UV280 or gel band intensity quantification? Given that the samples are in crude form without a purification step, they likely contain ample amounts of host cellular proteins in the lysate supernatant. Considering that the protein of interest originates from the same bacterial class, how did the authors account for or eliminate the potential contribution of the host's endogenous PAP-1 to the observed activity?
7. The PAP-1 enzyme from various E. coli strains was tested for activity using crude samples without purification. While the relevance of PAP-1 in mRNA vaccine production is mentioned, the practical implications are not fully explained. How does the optimized production protocol contribute to scaling for industrial use? Discussing biotechnological applications, challenges (e.g., cost, scalability, regulatory issues), and how the optimized process compares to commercially available PAP-1 would enhance the study's impact. It would be valuable to clarify the commercial production process and its relation to the findings.
8. The authors should specify the molecular weight of the protein of interest. Including a supplementary figure of SDS-PAGE gel showing the expression profile of the recombinant protein would be valuable. This would confirm the presence and relative abundance of the target protein in the lysate and enhance the reproducibility and clarity of the study.
9. To ensure accessibility for colorblind readers, I recommend avoiding color combinations that may be difficult to distinguish, such as red and green, or blue and purple. Consider using high-contrast color pairs (e.g., blue and orange, black and white) and incorporating patterns or symbols in the graphs to make the data more accessible to all readers.
Author Response
The manuscript titled "Optimization of Conditions for Production of Soluble E. coli PolyA-Polymerase" by Oscorbin et al. looks at how to improve the production of poly(A)-polymerase (PAP-1) in E. coli. This enzyme is essential for RNA polyadenylation and is used in RNA sequencing and mRNA vaccine development. The study compares the expression of PAP-1 in seven different E. coli strains to find the best conditions for producing the protein, making it soluble, and ensuring it works properly. The results emphasize how factors like the choice of E. coli strain, temperature during induction, and plasmid copy number are crucial for better protein production. These findings offer useful insights for improving PAP-1 production for use in biotechnology. While the experiment and results are solid, the manuscript could benefit from clearer statistical analysis and more discussion about how these results apply in real-world settings. I have a few comments and suggestions for improvement.
Dear reviewer, thank you for thoughtful comments and valuable suggestions! We greatly appreciate your efforts in reading and criticizing our manuscript. We carefully read and answered them point-by-point changing the manuscript in order to improve its clarity. All changes are highlighted by yellow color. Please find our answers to your notes below.
- The manuscript contains minor grammatical issues that require attention. For instance, in line 11, the phrase 'PAP-1's toxicity and instability complicate its expression and purification' should be corrected to 'complicate.' Additionally, there is a typographical error in line 131 ('poply(A)') and line 122 (1 mM ЭДТА). Subject-verb agreement errors are observed throughout the manuscript. It is recommended that the authors review the text carefully to ensure grammatical accuracy and clarity.
Thank you for your notion! We have corrected the mentioned typos and proof-read the manuscript in order to amend other similar issues. Hope that the quality of the manuscript was improved.
- The author thoroughly explained the method but did not include details regarding the sonication parameters used for cell lysis, such as duration, power settings, and on/off cycles. Were the same parameters applied to all samples? Was the lysis performed on individual samples or simultaneously? Given the significant impact of sonication parameters on cell lysis, protein folding, solubility, and activity, it would be helpful to clarify these aspects for a more comprehensive understanding of the experimental conditions.
Thank you for your comment! Indeed, sonication affects greatly on the efficacy of lysis and protein solubility. We have specified lysis and sonication parameters, please find below the paragraph added to the revised manuscript:
“All incubated probes were sonicated using a Cell Disruptor 200 (Branson Ultrasonics Corp., Brookfield, CT, USA) under the following conditions: 70 W for 1 minute in pulsed mode with a 30% duty cycle, repeated twice. Probes from a single strain were lysed and sonicated within the same experiment to minimize variations in lysis time between samples.”
- It is well-established, and the author has also mentioned in two instances in the manuscript, that Lemo21(DE3) is designed to modulate the expression levels of potentially toxic or difficult-to-express proteins using a tunable T7 RNA polymerase system controlled by the rhaBAD promoter, which is regulated by L-rhamnose. Therefore, did the authors add L-rhamnose during protein expression induction? If so, this should be specified in the methods section. If not, the absence of L-rhamnose could be a contributing factor to the insolubility of the expressed protein.
Thank you for your notion! We have not added L-rhamnose in the growth medium during our induction experiments with Lemo21 (DE3). This is one of limitations of our study, because fine tuning of the expression rate by adjusting L-rhamnose concentration may increase solubility of E. coli PAP-1. Indeed, too fast accumulation of PAP-1 in cells would lead to formation of inclusion bodies. Please find below the respective statement added in the Results section:
“A possible reason why PAP-1 was insoluble in Lemo21 (DE3) could be the rapid production of the recombinant protein, leading to its accumulation in inclusion bodies. This strain is designed for the tunable expression of recombinant proteins regulated by the L-rhamnose level. In this study, we did not adjust the expression of PAP-1 by adding L-rhamnose, which might have resulted in the rapid overproduction of PAP-1.”
- The experimental methods are detailed but could be improved for clarity and reproducibility. Specify whether conditions such as MgCl₂ or other cofactors (ATP etc.) were tested during expression, given the enzyme's reliance on divalent metals for activity.
Thank you for your comment! Here, we did not supply LB with any specific additives, such as MgCl2, ATP or other substances. Previously, many chemicals like ethanol, sorbitol, betaine, arginine were reported as improving overall yield or solubility of recombinant proteins. Physical methods, e.g., a short preheat before induction, were also applied for the same purposes. However, our main goal in the present study was to compare an ability of different E. coli strains to produce soluble and active E. coli PAP-1 which did not assumed usage of additional chemical substances in the cultural media. We have added the respective paragraph in the Discussion section, please find it below:
“A fruitful strategy to increase the yield, solubility, and activity of recombinant proteins in E. coli cells is supplementing the culture medium with various chemicals, such as ethanol (10.1016/j.mex.2015.09.005), sorbitol, betaine (10.1016/j.pep.2006.09.015), arginine (10.1128/AEM.05259-11), and glycerol (10.1006/mgme.2001.3172). These substances alter cellular metabolism and often increase the expression of chaperones, thus facilitating the synthesis of heterologous proteins. However, the main goal of this study was to compare the ability of different E. coli strains to produce soluble and active E. coli PAP-1, which did not assume the use of additional chemical substances in the culture medium.”
5. Some figures, such as the heatmap (Figure 2) and activity comparison (Figure 4), lack clear annotations and labels. For example, the color scale in the heatmap should be described in the legend, and key correlations should be highlighted. Additionally, Figure 4’s axes and legends could be better labeled to indicate the significance of the lysate volume adjustments for activity measurements.
Thank you for your suggestion! We have corrected Figures 2 and 4 in accordance with your recommendations by changing the style and legends of these figures. Other figure captions were also detailed. Please the changed figures and legends in the new version of the manuscript.
- In the experiment “Specific activity of E. coli PAP-1 (Figure 4),” calculating recombinant PAP-1 activity based on lysate volume, did the authors quantify the concentration of PAP-1 in each sample using methods such as UV280 or gel band intensity quantification? Given that the samples are in crude form without a purification step, they likely contain ample amounts of host cellular proteins in the lysate supernatant. Considering that the protein of interest originates from the same bacterial class, how did the authors account for or eliminate the potential contribution of the host's endogenous PAP-1 to the observed activity?
Thank you for your notion! It is a valid point that host E. coli cells also contain the same PAP-1 as was induced as the recombinant protein. Therefore, activity of endogenous PAP-1 would also be registered when we analyzed crude lysates with all host proteins. To rule out this possibility, we also analyzed PAP-1 activity in bacterial lysates from each E. coli strain without the pET-PAP-Eco plasmid using the same assay conditions as for the probes after induction. Thus, we estimated basal level of polyadenylation activity taking into account not only PAP-1 itself, but also PAP-2, PNPase also possessing the ability to polyadenylate RNA. Cells without pET-PAP-Eco were used as controls of basal polyadenylation level because in cells with the plasmid, recombinant PAP-1 will be inevitably produced due to leakage of the lac promoter. No elongation of (r)A20-oligonucleotide was registered with all control probes meaning low basal level of polyadenylation enzymes. Thus, host PAP-1, PNPase did not contribute to the substrate elongation in our activity assays and did not distort results of recombinant PAP-1 activity assay. To illustrate these observations, we added a Supplementary Figure 2 in the manuscript with the respective comments in the Results section. Please find them below:
“E. coli cells possess at least three enzymes capable of polyadenylating RNA: PAP-1, PAP-2, and PNPase. The activity of these enzymes in crude lysates after recombinant PAP-1 expression could distort the results of polyadenylation assays. To account for host polyadenylation activity, we used control cultures of all strains that did not contain the pET-PAP-Eco plasmid, which could produce recombinant PAP-1 due to promoter leakage. All control cultures were prepared identically to those expressing recombinant PAP-1. No elongation of the (r)A20-oligonucleotide was observed with any control probes, indicating a low basal level of polyadenylation enzymes (an example is provided in Supplementary Figure 2). Therefore, host enzymes did not distort the results of polyadenylation activity in crude lysates.”
- The PAP-1 enzyme from various E. coli strains was tested for activity using crude samples without purification. While the relevance of PAP-1 in mRNA vaccine production is mentioned, the practical implications are not fully explained. How does the optimized production protocol contribute to scaling for industrial use? Discussing biotechnological applications, challenges (e.g., cost, scalability, regulatory issues), and how the optimized process compares to commercially available PAP-1 would enhance the study's impact. It would be valuable to clarify the commercial production process and its relation to the findings.
Thank you for your recommendation! The information about possible implementation of the presented results into industrial processes was added in the Discussion section of corrected manuscript. Specifically, we have stressed out that the host strain affects greatly on the efficacy of PAP-1 production, and proprietary E. coli strains not always are the best possible choice for that task. Benefits and limitations of other methods to increase recombinant protein production are also mentioned, including co-expression with heat shock proteins and supplying of cultural media with various chemicals. Please find below the added paragraph.
“Our results demonstrated that the host strain has a significant impact on the efficacy of PAP-1 production, and proprietary E. coli strains are not always the best choice for this task. While seemingly time-consuming and cumbersome, systematic preliminary experiments can save both funds and time if they demonstrate that freely available E. coli strains express the same amount of recombinant protein as commercial strains. It should be noted that further optimization, specifically co-expression with heat shock proteins and supplementation of the culture medium with various chemicals, could further increase PAP-1 production, as shown for other proteins. However, in large-scale production, additional chemicals can unexpectedly increase the cost of fermentation, not only because of the cost of the substances themselves but also due to the need for proper waste disposal. The use of heat shock proteins may also be limited by their negative impact on E. coli propagation as an additional physiological burden or by their cost if commercially available proteins are used. Thus, the resulting production cost needs to be carefully considered before implementing any additives into the manufacturing process. In this light, strain optimization appears to be one of the most economical approaches, as it only needs to be done once and does not require additional expenditures if a freely available strain demonstrates an acceptable production rate.”
- The authors should specify the molecular weight of the protein of interest. Including a supplementary figure of SDS-PAGE gel showing the expression profile of the recombinant protein would be valuable. This would confirm the presence and relative abundance of the target protein in the lysate and enhance the reproducibility and clarity of the study.
Thank you for your suggestion! Indeed, gels with results of expression experiments are necessary in studies of this kind and should be provided to demonstrate the actual presence of the protein of interest in analyzed probes. Please find the example of gels as Supplementary Figures 1.
- To ensure accessibility for colorblind readers, I recommend avoiding color combinations that may be difficult to distinguish, such as red and green, or blue and purple. Consider using high-contrast color pairs (e.g., blue and orange, black and white) and incorporating patterns or symbols in the graphs to make the data more accessible to all readers.
Thank you for your suggestion! We have changed all figures in accordance with your advice to make them more readable for color-blind people. Specifically, color patter is changed to the scheme proposed by Bang Wong (10.1038/nmeth.1618). Bar patterns are also added to make the “coding” redundant to facilitate the distinguishing of the presented data.
Reviewer 2 Report
Comments and Suggestions for Authors
The manuscript is well-written and clearly organized. The writing is clear and concise. The goal of this study is clearly stated. The following are my suggestions for this manuscript. The title of the manuscript could be changed a little by adding ‘ …..for biotechnological applications’ in the end. The significance of this manuscript is missing in the title. The simple summary and abstract are almost the same. The introduction could have ended with a brief future prospective of this study. Also, the conclusion could have been concluded with future prospective of this manuscript. Any suggestions on future improvements or next steps in the expression and purification of PAP? The figures need formatting, and the legends could have been more in detail. But the overall figures are nicely represented with colors so that it becomes self-explanatory. Are there any ethical concerns regarding the use of genetically modified organisms in this study? Overall, very interesting manuscript to read.
Here are my additional comments. • What is the main question addressed by the research? The main question addressed by the research is to determine the optimal conditions for the production of valuable recombinant soluble E.coli PAP-1 which will be further used in biotechnological applications or mRNA vaccine use.
• Do you consider the topic original or relevant to the field? Does it
address a specific gap in the field? Please also explain why this is/ is not
the case. Yes, this topic is original because the group has studied seven strains of E.coli for their overall protein yield, solubility, and enzymatic activity and determined the optimized conditions for BL21 (DE3) pLysS strain.
• What does it add to the subject area compared with other published
material? Yes, this group has tested several strains of bacteria with different conditions when it is compared with other published material.
• Are the conclusions consistent with the evidence and arguments presented
and do they address the main question posed? Please also explain why this
is/is not the case. Yes, the conclusions are consistent with the evidence and arguments made before and address the main question posted.
• Are the references appropriate?Yes
• Any additional comments on the tables and figures. Certain formatting is needed in the figures. The figures are overall well presented and self-explanatory.
Author Response
The manuscript is well-written and clearly organized. The writing is clear and concise. The goal of this study is clearly stated. The following are my suggestions for this manuscript. The title of the manuscript could be changed a little by adding ‘ …..for biotechnological applications’ in the end. The significance of this manuscript is missing in the title. The simple summary and abstract are almost the same. The introduction could have ended with a brief future prospective of this study. Also, the conclusion could have been concluded with future prospective of this manuscript. Any suggestions on future improvements or next steps in the expression and purification of PAP? The figures need formatting, and the legends could have been more in detail. But the overall figures are nicely represented with colors so that it becomes self-explanatory. Are there any ethical concerns regarding the use of genetically modified organisms in this study? Overall, very interesting manuscript to read.
Dear reviewer, thank you for your efforts in reviewing of our manuscript! We greatly appreciate your helpful advices and constructive recommendations. We carefully read and answered them consistently changing the manuscript in order to improve its clarity and quality. All changes are highlighted by yellow color. Please find below our answers to your notes.
- The title of the manuscript could be changed a little by adding ‘ …..for biotechnological applications’ in the end.
Thank you for your suggestion! We have changed the title in accordance with your recommendation, as it will clearer demonstrate the topic of the study.
- The simple summary and abstract are almost the same.
Thank you for your notion! Indeed, abstract and the simple summary are almost identical. Writing the abstract, we tried to make it as clean and readable as possible for the broader audience that can be unfamiliar with specific terms. In that sense, further simplifying of the abstract would lead to loss of the important information. Based on this assumption, we decided not abridge the summary.
- The introduction could have ended with a brief future prospective of this study. Also, the conclusion could have been concluded with future prospective of this manuscript. Any suggestions on future improvements or next steps in the expression and purification of PAP?
Thank you for your recommendation! We have added in the end of the Introduction and Discussion sections our view on possible future developments on the expression and purification of E. coli PAP-1. Specifically, we listed several methods for increasing PAP-1 solubility and yield: the usage of helper proteins, e.g., molecular chaperones; supplying of culture medium with additives such as sorbitol, ethanol, betaine, L-arginine; short temperature shock before addition of IPTG to induce synthesis of host heat shock proteins. Lysis conditions, such as ionic strength, detergents and chaotropic agents are also mentioned. Please find below the added paragraphs.
Introduction:
“In the future, the production and solubility of PAP-1 can be further improved using several methods: co-expression with molecular chaperones; supplementation of the culture medium with additives such as sorbitol, ethanol, betaine, and L-arginine; a short temperature shock before the addition of IPTG to induce the synthesis of host heat shock proteins; and optimization of lysis conditions, such as ionic strength, detergents, and chaotropic agents.”
Discussion:
“Here, we focused solely on comparing different E. coli strains for the production of recombinant E. coli PAP-1. Other methods commonly used to increase the yield and solubility of recombinant proteins, such as co-expression with molecular chaperones, supplementation of the culture medium with additives, induction of host heat shock proteins, and optimization of lysis conditions (e.g., ionic strength, detergents, and chaotropic agents), could also be beneficial but remain subjects for possible future studies.”
- The figures need formatting, and the legends could have been more in detail. But the overall figures are nicely represented with colors so that it becomes self-explanatory.
Thank you for your suggestion! We have changed figures and their legends to make them more readable and clarify methodological details. Specifically, colors were changed in order to facilitate their recognition by color-blind people.
- Are there any ethical concerns regarding the use of genetically modified organisms in this study?
Thank you for your notion! Here, we used only standard genetic manipulations with E. coli cells, e.g. transformed them with a plasmid carrying a PAP-1 gene. No modifications were introduced in E. coli genome. Therefore, we assumed that no specific statement is necessary about the usage of GMO in our study, as we only worked with a plasmid without any manipulations with E. coli genomic DNA.
Round 2
Reviewer 1 Report
Comments and Suggestions for Authors
The authors have addressed the comments and concerns adequately. The revision has improved the manuscript's quality and clarity. The revised manuscript is acceptable to me.